# SERPINA3: Stimulator or Inhibitor of Pathological Changes

**DOI:** 10.3390/biomedicines11010156

**Published:** 2023-01-07

**Authors:** Mateusz de Mezer, Jan Rogaliński, Stanisław Przewoźny, Michał Chojnicki, Leszek Niepolski, Magdalena Sobieska, Agnieszka Przystańska

**Affiliations:** 1Department of Immunobiology, Poznan University of Medical Sciences, Rokietnicka 8, 60-806 Poznan, Poland; 2Department of Physiology, Poznan University of Medical Sciences, Swiecickiego 6, 61-781 Poznan, Poland; 3Department of Physiotherapy, Chair for Rehabilitation and Physiotherapy, Poznan University of Medical Sciences, 28 Czerwca 1956 r. 135/147, 61-545 Poznan, Poland; 4Department of Anatomy, Poznan University of Medical Sciences, Swiecickiego 6, 61-781 Poznan, Poland

**Keywords:** SERPINA3, α-1-antichymotrypsin, anti-inflammatory, antiapoptotic, DNA binding, PI3K/AKT, MAPK/ERK 1/2

## Abstract

SERPINA3, also called α-1-antichymotrypsin (AACT, ACT), is one of the inhibitors of serine proteases, one of which is cathepsin G. As an acute-phase protein secreted into the plasma by liver cells, it plays an important role in the anti-inflammatory response and antiviral response. Elevated levels of SERPINA3 have been observed in heart failure and neurological diseases such as Alzheimer’s disease or Creutzfeldt–Jakob disease. Many studies have shown increased expression levels of the *SERPINA3* gene in various types of cancer, such as glioblastoma, colorectal cancer, endometrial cancer, breast cancer, or melanoma. In this case, the SERPINA3 protein is associated with an antiapoptotic function implemented by adjusting the PI3K/AKT or MAPK/ERK 1/2 signal pathways. However, the functions of the SERPINA3 protein are still only partially understood, mainly in the context of cancerogenesis, so it seems necessary to summarize the available information and describe its mechanism of action. In particular, we sought to amass the existing body of research focusing on the description of the underlying mechanisms of various diseases not related to cancer. Our goal was to present an overview of the correct function of SERPINA3 as part of the defense system, which unfortunately easily becomes the “Fifth Column” and begins to support processes of destruction.

## 1. Introduction

SERPINA3, also called α-1-antichymotrypsin (AACT, ACT), is a protein that belongs to the family of protease inhibitors. It is coded by the serine protease inhibitor A3 (*SERPINA3*) gene, mapped on the 14q32.13 region of chromosome 14 [1,2] near other serpin genes (Figure 1A). *SERPINA3* is up to 11,5 kbp long and has five exons coding for three different mRNAs, containing the same 1271 nt coding sequence (CDS) (NM_001085.5) (Figure 1A). Its translation results in the SERPINA3 protein, built by 423 amino acids with a molecular mass of 46 kDa containing a DNA-binding domain and reactive center loop (RCL) (Figure 1B). Due to later post-translational modifications at six sites of glycosylation [3,4], its molecular mass increases by approximately 25% to 55–66 kDa [5,6]. However, recent data show that SERPINA3 mRNA is subject to alternative splicing. In one of the possibilities, exon E1 is replaced by exon N1, which leads to the formation of a protein with an altered N-terminal sequence. In turn, changing the splicing site at the end of exon E3 introduces an additional 12 nt containing the stop codon, which leads to the formation of a protein without the RCL sequence encoded by exon E4 (Figure 1C) [7]. The form called ACT-N appears to be catalytically active but may show reduced substrate binding capacity. The lack of the N-terminal part, which is considered by some authors to be a signal peptide, may affect the location of this form of the SERPINA3 protein. In contrast, the ACT-T form does not have protease inhibitor activity [7]. Thus far, the functions of the ACT-N and ACT-T forms have not been defined.

SERPINA3 belongs to the α1-globulin fraction of serum proteins, and its gene expression is mainly regulated by IL1 and IL6 cytokines via the STAT3 (signal transducer and activator of transcription 3) pathway [8,9]. It is mainly produced by the liver and secreted by hepatocytes into peripheral blood to its final concentration of 0.35–0.45 g/L in a healthy individual [10]. Together with other liver-specific proteins, the expression of SERPINA3 starts noticeably early (on the 18th day) during embryonal development, after the induction of hepatocytes by oncostatin M (OSM) [11].

The SERPINA3 protein is also physiologically present in the gallbladder, the pancreas, the prostate, the testes, the uterus [12], and the brain, where it is produced by astrocytes [13]. It can also be synthesized by keratinocytes in response to local skin damage [8]. It is worth noting that the tissue expression of the *SERPINA3* gene is differentiated at both the RNA and protein levels (Figure 2). The fact that the liver produces SERPINA3, which is primarily a secreted protein, is evidenced by a high level of mRNA at an average protein level. High levels of the SERPINA3 protein are observed within the uterus (https://www.proteinatlas.org/ENSG00000196136-SERPINA3/tissue, accessed on 17 December 2022), possibly associated with the intensive changes that occur in the tissue structure during physiological processes in the uterus. SERPINA3 is categorized as an acute-phase inflammatory reaction protein because, when stimulated by cytokines, its serum concentration increases 2–5 times during immune responses [2]. Interestingly, SERPINA3 mRNA is undetected in the blood [14].

The main function of SERPINA3 is the inhibition of serine proteases, such as chymotrypsins, cathepsin G, and mast cell chymase, by binding them in a stable complex, which prevents them from the proteolytic activity and consequently leads to changes in the extracellular matrix (ECM) [15,16,17]. The active site of enzyme reactivity is localized between Leu358 and Ser359 [2].

## 2. SERPINA3 as a Nuclear Protein

The function of the SERPINA3 protein is more complicated than previously believed due to the unique presence of a double-stranded DNA-binding domain (formed by repetitive lysin motives in the 210–212 and 235–237 protein regions) (Figure 1B). Interestingly, such protein–DNA interactions do not affect the inhibitory activity of serine proteases [18].

Santamaria et al. observed that the transportation of the N-glycosylated form of the SERPINA3 protein (fraction 55 kDa) into the cellular nucleus depends on its interaction with importin α/β protein. There, bonded to DNA, SERPINA3 causes chromatin condensation, which occurs in large nuclear complexes. SERPINA3 seems to inhibit DNA polymerase and consequently decreases DNA synthesis. They observed that during mitosis, the overexpression of AACT causes the accumulation of cells in the G0/G1 phase [11]. SERPINA3 can inhibit cellular growth, proliferation, and differentiation owing to its nuclear localization.

Furthermore, in some tumors, the SERPINA3 protein shows antiapoptotic activity by the inhibition of normal cellular transfer from the G2 phase to the M phase by activating the MAPK/ERK1/2 and PI3K/AKT pathways. The increased level of SERPINA3 in breast, colon, and prostate cancers, as well as glioma, proves that theory [19,20,21,22].

## 3. SERPINA3 in Cancers

The SERPINA3 protein plays a vital role in tumorigenesis, and its increased level is associated with worse prognosis in some cancers; therefore, it has thus far been used as a diagnostic factor in colon, breast, lung, and gastric cancers.

SERPINA3 promotes tumor development by regulating the transcription of some oncogenes, for example in hepatocellular carcinoma, by increasing the length of telomeres, cell proliferation, and migration [23]. In this case, the transcriptional activity of SERPINA3 is modulated by reactive oxygen species (ROS), which activates the PI3κδ pathway [24].

The tumorigenic function of *SERPINA3* is also due to its genetic polymorphism. For example, an abnormal change in genetic sequence 918G>C increases the risk of developing breast cancer by three times, as well as medullary thyroid tumor (especially in BRCA1 genetic carriers) [25]. Koivuluoma et al. showed that the overexpression of *SERPINA3*, induced by estrogens in patients with positive ER/PR breast cancer, could be used as a prognostic factor [25]. Zhang et al. proved that an overexpressed *SERPINA3* gene stimulates the proliferation, invasion, and migration of triple-negative breast tumor (TNBC) cells by causing the overexpression of the enhancer of zeste homolog 2 (EZH2), which, in turn, promotes epithelial–mesenchymal transition (EMT) and cellular differentiation [26]. Thus far, EZH2 is the only known factor; therefore, further research must be carried out to exclude other mechanisms. Additionally, the authors noted that raised SERPINA3 protein levels are correlated with the resistance of TNBC to cisplatin treatment; however, the actual mechanism is still unknown [26].

The overexpression of *SERPINA3* is also positively correlated with glioma development, as well as its size, stage (WHO grade), and negative prognosis. The presence of the SERPINA3 protein promotes the remodeling of astroglia and the extracellular matrix of microglia [12,27,28]. Nimbalkar et al. showed that SERPINA3 may contribute to glioma cell proliferation, invasion, or migration [29,30]. Interestingly, SERPINA3 acts on both cancer cells and immune cells by inhibiting the immune response to tumors. The level of SERPINA3 protein correlates with low infiltration of glioma tissue by CD4+ cells. The expression level of SERPINA3 is reversely correlated with the number of M1 macrophages, monocytes, and activated mast cells [31].

Patients with ulcerative colitis (UC) have elevated expression of SERPINA3 mRNA and its protein in diseased tissues [32]. UC is characterized by severe inflammation. SERPINA3 protein levels are known to be positively correlated with the levels of cytokines such as IL1β, IL6, IL8 (CXCL8), and TNFα on HT29 intestinal epithelial cells. This relationship was also observed when a knockdown of the SERPINA3 protein was induced. According to the authors, these findings point to both the pro-inflammatory role of SERPINA3, which appears to act as a factor stimulating gene expression, and to its usefulness as a diagnostic marker readily available for analysis in blood and urine [32]. At the same time, UC is one of the risk factors for colorectal carcinoma (CRC) cancer, which has been confirmed in studies in which 71% of CRC cases have been diagnosed with ulcerative colitis [33]. As CRC develops, higher levels of the SERPINA3 protein are observed [34,35]. It has also been noted that high levels of the SERPINA3 protein in tumors are correlated with poor cell differentiation, increased vascularity, and the occurrence of liver metastases [34]. At the same time, a stimulating effect of the SERPINA3 protein on the phosphorylation of AKT and ribosomal protein S6 resulting in an increase in the proliferation of HT-29 colon cancer cells was observed [36]. The fact is that inflammation is strongly associated with both ulcerative colitis and carcinogenesis, so it is necessary to conduct further research on whether the SERPINA3 protein is specifically associated with CRC or only with the inflammatory pathway as an acute-phase protein.

SERPINA3 protein levels may also be upregulated in the plasma of patients with squamous cell esophageal cancer [37] and in the cerebrospinal fluid (CSF) in patients suffering from extranodal NK/T-cell lymphoma [38]. However, the relevance of these observations for the course of the illness or diagnostic or therapeutic processes has not yet been determined. Unexpectedly, a decrease in the levels of both the SERPINA3 protein and its GlcNAcylated form in the early stages of non-small-cell lung cancer was observed; however, in the subsequent stages of the disease, levels increase and may be very high at a late stage, probably due to its role in tumor development [39].

Elevated *SERPINA3* expression was also revealed in cells of primary melanoma as well as its metastasis. Kulesza et al. studied STAT3-dependent genes in melanomas and found that STAT3 binds to the promoter of the *SERPINA3* gene, causing its activation. *SERPINA3* knockdown was shown to inhibit the invasion and migration of melanoma cells [40,41]. Interestingly, STAT3 is one of the transcription factors responsible for the increase in *SERPINA3* gene expression after the stimulation of the integrin–FAK–AKT signaling pathway, which leads to the migration of melanoma cells [42]. It appears that higher expression of the SERPINA3 protein may cause more aggressive tumor biology due to the remodeling of the extracellular tissue matrix.

High expression of the *SERPINA3* gene is also associated with poor prognosis, low differentiation, malignancy, and high-stage endometrial cancer. Interestingly, high expression of *SERPINA3* was observed in ER-negative expression cells [43], which may indicate the existence of common regulatory elements, the understanding of which could shed new light on the development of anticancer therapy. It has been observed that silencing the expression of the *SERPINA3* gene leads to the inhibition of proliferation of cancer cells by stopping the cell cycle in the G2/M phase, which consequently leads to apoptosis. The SERPINA3 protein is likely involved in the activation of PI3K/AKT and MAPK [43,44].

Observation of the elevated levels of the SERPINA3 protein in solid cancerous tumors prompts the search for elements that can regulate the expression of the *SERPINA3* gene in these tissues. For more detailed information about the cell signaling pathway necessary for the regulation of *SERPINA3* gene expression, we refer the reader to a study by Soman and Nair entitled “Unfolding the cascade of SERPINA3: Inflammation to cancer”, which thoroughly describes the role of the SERPINA3 protein in cancer by presenting highly accurate data [45].

## 4. The Role of Non-Coding RNA in the Regulation of *SERPINA3* Gene Expression and Function

It has been approximately twelve years since the modification of “the central dogma of molecular biology” formulated by Francis Crick in 1957 [46]. The main reason for the change in the way we understand the processes leading to the expression of genetic information is the progress in understanding the sequence and cellular role of non-coding RNAs (ncRNAs). This is a fraction of cellular RNAs that are not directly related to protein synthesis but act at the level of interaction with mRNA. The ncRNAs are divided into groups differing in size and mechanisms of action. Thus, short-chain ncRNAs include microRNAs, while ncRNAs with molecules longer than 200 nt are called long non-coding RNAs (lncRNAs). Round RNAs (circRNAs) are a special class of ncRNAs that differ from linear RNAs in that they have a closed, circular form [47]. Most of the knowledge about ncRNAs is derived from the studies of tumorigenesis but we sought to relate these data to other processes in which the SERPINA3 protein is involved. Circular RNA (circRNA) is one of the key elements of regulation in eukaryotic transcriptomes, and often exhibits tissue- and developmental-specific expression [48]. Analysis of prostate cancer showed that the circSERPINA3 level is elevated in cells of this tumor [49]. Studies have shown that circSERPINA3 reduces the half-life of the *SERPINA3* gene mRNA in cells by competitively binding the RNA-binding protein BUD13 and miR-653-5p. It has been shown that the key role in this process is played by blocking the interaction of the BUD13 protein with SERPINA3 mRNA. By lowering the expression of the *SERPINA3* gene, circSERPINA3 can promote autophagy and aerobic glycolysis [50]. At the same time, Liu et al. showed that circSERPINA3 promotes the progression of nasopharyngeal cancer (NPC) by capturing miR-944 and thus increasing the level of the MDM2 protein, which induces cell proliferation and invasion. An increase in circSERPINA3 levels was also observed with the worse overall survival of patients with NPCs [51].

On the other hand, based on miRDB analysis (http://mirdb.org/index.html, accessed 13 May 2022), only two miRNAs are predicted to be matched to SERPINA3 mRNA. These are has-miR-296-5p (target score 63) and has-miR-137-3p (target score 53).

The miR-137 may inhibit the expression of extracellular matrix proteins and promote cell growth [52] and, as a regulator of cellular differentiation and cell cycle control, is involved in cancer invasion [53]. It has been confirmed that miR-137 can inhibit *SERPINA3* expression [54]. Chen et al. described the suppression of miR-137 expression by LncRNA GAS5 [55] and suggested that the LncRNA GAS5/miR-137/SERPINA3 axis (Figure 3) may play an important role in some pathological conditions, such as myocardial damage. Using a rat model, the authors showed that inhibition of GAS5 leads to a decrease in the apoptosis of myocardial cells, fibrosis, and pathological injuries. However, the thesis that the LncRNA GAS5/miR-137/SERPINA3 axis may play a key role in myocardial damage requires experimental evidence [56]. It was confirmed that during cardiac arrest/cardiopulmonary resuscitation (CA/CPR), the binding of miR-137 by GAS5 increases the expression of inositol polyphosphate-4-phosphatase type II B (INPP4B). INPP4B causes the selective degradation of PIP2 and PIP3 proteins, which leads to the inhibition of PI3K/Akt signaling activation [57]. In this way, there may also be a modification of the action of the SERPINA3 protein, which, as mentioned earlier, seems to be one of the activators of AKT; alternately, there may be an imbalance between the activating action of SERPINA3 and the inhibitory effect of INPP4B.

In turn, miR-296 is involved in the regulation of cell proliferation and the secretion of pulmonary surfactants [58], whereas in cervical cancer (CC) the inhibition of miR-296-5p by circular RNA E2F transcription factor 3 (circ-E2F3) promotes the nuclear translocation of STAT3, increasing the proliferation and migration of CC cells [59]. In this case, we also believe an increase in the expression of the SERPINA3 protein may occur as a result of the increased effect of STAT3 at the promoter level and a reduction in mRNA degradation by miR-296.

## 5. The Role of SERPINA3 in Inflammation

As mentioned earlier, SERPINA3 is classified as a protein in the acute phase of inflammation. During the acute-phase response to emerging inflammation, an increase in the plasma concentration of the SERPINA3 protein is observed, similar to changes in the concentration of C-reactive protein (CRP). This indicates the potential contribution of this protein to the control of inflammation, especially if no massive tissue damage has occurred, as the concentration of the SERPINA3 protein relatively quickly returns to baseline values. The alternative name α-1-antichymotrypsin indicates the primary function of this protein. The task of antiproteases is to prevent tissue damage during diapedesis and phagocytosis caused by neutrophils in damaged tissue. It should be noted, however, that SERPINA3 is not the main antiprotease and seems to play another, as yet undefined, role [60].

It has been observed that the plasma concentration of the SERPINA3 protein increases with the prolongation of inflammation and accompanies multiorgan injuries or burns. The longer the inflammatory process, the more pronounced the increase in the plasma levels of SERPINA3 [61]. Moreover, the increase in concentration is proportional to the area of necrosis, and when necrosis stops progressing, the concentration of SERPINA3 decreases to a level slightly above the baseline [62,63]. The foci of necrosis have been confirmed in the removed tonsils, suggesting that the necrosis of large lymph nodes may also be associated with increased concentrations of the SERPINA3 protein [61,64]. SERPINA3 is, therefore, not so much a marker of the presence of the inflammatory process itself, as it may indicate its particularly destructive nature [65].

During inflammation, besides the concentration of the SERPINA3 protein, the profile of its glycosylation also changes. Based on its reaction with concanavalin A, the glycosylation formula of the SERPINA3 protein was described. Four variants (A1 to A4) were found to exist in healthy individuals in the proportions of 21:32:28:19%. In addition, the physiological presence of small amounts (not exceeding 5%) of variant A5 was noted in children. This variant in adults appears only in states of the strongest inflammatory response, caused, for example, by multiorgan trauma, and the amount can be up to 20% of the total plasma pool of the SERPINA3 protein [66]. Interestingly, a significant change in the glycosylation profile was observed in isolated head trauma, even at its low severity. In sepsis, changes in the SERPINA3 protein glycosylation are reversed [63].

It is widely recognized that glycosylation is one of the most important modifications that proteins undergo in their maturation process. It is believed that these modifications affect the stability of the tertiary structure of proteins and their susceptibility to proteolysis. Protein glycosylation has been shown to play a key role in the regulation of several processes such as cellular adhesion, cell differentiation, and cell-to-cell communication. It is also known that its disorders lead to several human diseases [67,68]. The research results described above indicate that in the SERPINA3 protein this process not only involves the “finish” of the peptide but above all also allows this protein to change its mode of action. Unfortunately, the underlying mechanisms that regulate the changes in SERPINA3 glycosylation have not yet been described, and there is a lack of information about their importance for the functioning of this protein. Therefore, this poses a huge challenge for researchers, especially since the presented observations indicate that the monitoring of protein levels is an important element of the diagnostic process during convalescence, not only after extensive injuries but also in the case of seemingly small procedures such as tonsillectomy. The monitoring of SERPINA3 protein glycosylation in routine diagnostics could allow for much earlier detection of sepsis development and perhaps more effectively inhibit it.

## 6. SERPINA3 in Antiviral Response

In addition to injuries and inflammation, viral infections trigger the activation of the body’s defense mechanisms. In this direction, to study the processes induced by the appearance of viruses in the body, changes in the level of the SERPINA3 protein have been investigated. In their study, Burgener et al. analyzed the changes in protein expression in the cells of HIV-1-infected, HIV-1-uninfected, and HIV-1-resistant individuals [69]. The level of the SERPINA3 protein was significantly higher in HIV-1-resistant subjects with no increase in cytokines (IL-1α, IL1-β, IL-6, IL-8, and TNF-α), which confirmed the lack of association between an increase in *SERPINA3* gene expression and the inflammation state in this situation. At the same time, studies on human brain microvascular endothelial cells (HBMEC) showed an increase in the level of the SERPINA3 protein under the influence of the viral TAT protein of subtype B (Tat.B) [70]. These observations allow us to associate the increase in SERPINA3 levels with the stimulation by Tat.B of the NF-kB pathway [71], leading to the activation of MAPK kinases. In addition, Chasman et al. suggested the action of the SERPINA3 protein as an inhibitor of viral replication based on the observation of an increase in the amount of virus in cells in which *SERPINA3* gene expression was reduced [72]. Similar studies using cell lines were conducted by Ferrarini et al., who observed an increase in the level of SERPINA3 mRNA in the cells infected with the SARS-CoV2 virus [73]. Interestingly, according to the authors, among all the cell lines used this is the only immunomodulatory factor whose expression was stimulated by a viral infection. Plasma analysis of the patients infected with SARS-CoV2 also showed that SERPINA3 protein levels were higher in patients with increased symptoms than in patients with milder infections [74]. These studies are consistent with those presented by Nunez et. al., who showed that SARS-CoV2 patients had elevated levels of the plasma protein SERPINA3, persisting even after symptoms resolved and hospitalization ended [75]. Akgun et al. found that one of the proteins in greater quantities in SARS-CoV2 (+) swabs is cathepsin G [76]; however, given the inhibitory effect of the SERPINA3 protein on cathepsin G, this is a surprising observation.

Abbasi et al. focused on the expression of genes involved in immunity, inflammation, and antiviral responses in cases of severe acute respiratory infection (SARI) caused by the human rhinovirus (HRV) among children aged 0–5 years [77]. They showed that SERPINA3 mRNA levels were elevated in cases of HRV infection.

However, the presented observations of the increase in the expression of the *SERPINA3* gene during viral infections indicate its significant participation in this process. It is, therefore, necessary to conduct further studies that will determine the function of the SERPINA3 protein in response to infection and its mechanism of action. This search seems all the more justified because the SERPINA3 protein has not yet been classified as a damage-associated molecular pattern (DAMP) [78], although it appears to meet the conditions set out in the definition of “Any molecule which is exposed during, after, or because of disrupted cellular homeostasis such as damage or injury” [79].

## 7. SERPINA3 in Heart Failure

Changes in the SERPINA3 protein levels in chronic and acute heart disease have been well documented and studied in recent years. Jiang et al. identified the SERPINA3 protein as a potential biomarker in heart failure based on an analysis of the Gene Expression Omnibus (GEO) database, which contains information about mRNA levels in heart failure (HF) [80]. Differentially expressed genes (DEGs) were analyzed and functional correlation analyses were performed. This previous study confirmed the role of immune infiltration in the progression of myocardial fibrosis. A correlation between the highly expressed *SERPINA3* gene and xenobiotic metabolism, inflammatory response, and adipogenesis was also described. The results suggest that the determination of the level of *SERPINA3* gene expression could be used to prevent and monitor the treatment of heart failure after myocardial infarction [80]. This finding is even more validated because Delrue et al. showed that the level of circulating SERPINA3 protein was a clear prognostic indicator in patients with de novo or worsening heart failure [81]. The study was based on gene transcripts and analysis of the proteins from the left ventricle of the heart of patients who died from HF. The authors suggest that the circulating SERPINA3 protein originates from vascular endothelial cells, where it is produced in response to inflammatory cytokines. It has also been shown that a greater amount of the SERPINA3 protein in circulation can inhibit the accumulation of neutrophils in ischemic and reperfused myocardium and can also inactivate the cytotoxic metabolites released from neutrophils. The observed elevated level of the SERPINA3 protein in atherosclerosis may be associated with the remodeling of atherosclerosis plaques and their stability, which increases the risk of a heart attack. Plasma SERPINA3 protein fraction was, therefore, indicated as a candidate for a plasma biomarker differentiating between myocardial damage and stable angina. Its amount is also increased in acute myocardial damage (AMI), possibly caused by angiotensin II and cytokine storms [82].

## 8. SERPINA3 in Neurological Diseases

Extensive research has been performed on the role of SERPINA3 in pathological changes in Alzheimer’s disease (AD). Elevated levels of the SERPINA3 protein have been found in the blood and the brain, including the hippocampus, as well as in cerebrospinal fluid (CFS) [83,84,85,86]. Analysis of the components of the senile plate showed that next to the peptide β-amyloid (Ab), the SERPINA3 protein is their main component [87,88]. Initially, it was found that both active and proteolytically fission forms of the SERPINA3 protein are present there [89]. Interestingly, the Ab/SERPINA3 complex activates the expression of transcription factors PPARγ and NF-κβ [13]. In light of the aforementioned observations suggesting the involvement of SERPRINA3 in the induction of the NF-κβ pathway in cooperation with the viral protein Tat.B, it seems reasonable to conclude that SERPINA3 can act as a transcription factor regulating the expression of NF-κβ, the activity of which is modified by interaction with other proteins. On the other hand, animal studies have shown that apoE4 can raise Serpina3n levels in the brain of mice [90]. It should be noted that homology is problematic because the HomoloGene database (https://www.ncbi.nlm.nih.gov/homologene, accessed on 16 December 2022) shows four mouse genes indicated as the orthologs of human *SERPINA3*. The most commonly used functional *Serpina3n* with a similar role is human *SERPINA3*, with which it has 61% homology [31], and the other genes are *Serpina3c*, *Serpina3m*, and *Serpina3k*. Due to such incomplete similarity of the sequence of the human *SERPINA3* gene and mouse genes as well as the lack of unambiguous data on the function performed by the mouse protein, it is necessary to interpret data obtained using the animal model with great caution.

The SERPINA3 protein fraction found in amyloid plaques is rather produced by the astrocytes surrounding Ab deposits rather than derived from the blood, where it could cross the damaged blood–brain barrier (BBB) via the CFS [91]. At the same time, it has been observed that increasing levels of the SERPINA3 protein in the CFS may be a marker of amnestic mild cognitive impairment (MCI) during AD progression [91]. The important role of the *SERPINA3* gene in the pathogenesis of AD is confirmed by the observation of an association between the polymorphism of codon-17 (A > T) in the SERPINA3 promoter region and the earlier age of the onset of disease symptoms [92].

Interestingly, the increasing levels of the SERPINA3 protein were also reported in the CFS collected by a lumbar puncture in patients with multiple sclerosis (MS). Its highest levels in the CFS are observed in patients with progressive MS, both primary progressive MS (PPMS) and secondary progressive MS (SPMS) [93]. Studies of the animal model of MS and the results of analyses of the material obtained from patients suggest that the determination of SERPINA3 protein levels in the CSF may be a useful marker of disease progression and neurodegeneration. Notably, the variation in the level of SERPINA3 in the blood in this case has not been measured, so its diagnostic usefulness cannot be determined.

The search for the genetic causes of schizophrenia shows that the expression of two genes, known as “neuroinflammatory” genes, is elevated. These are human immunodeficiency virus enhancer binding protein 2 (HIVEP2) and SERPINA3. Both depend on the transcription factor NF-κβ, but changes in the level of the SERPINA3 protein in astrocytes are not associated with fluctuations in the level of the HIVEP2 protein. Studies using a mouse model of schizophrenia have shown that in this type of pathology the SERPINA3 protein is produced by both astrocytes and neurons under the control of NF-κβ. Astrocytes with elevated expression of the *SERPINA3* gene were observed at perivascular localization, which may suggest a possible role for the SERPINA3 protein in promoting macrophage migration across the BBB [94].

An important group of neurodegenerative disorders is prion diseases, which include Creutzfeldt–Jakob disease (CJD). Vanni et al. identified several genes with variable expression in the frontal cortex in patients affected by prion disease [14]. They observed that in iatrogenic CJD samples, the mRNA level of the *SERPINA3* gene is significantly elevated, reaching 350 times the normal level. At the same time, they found a proportional increase in protein levels in both astrocytes and neurons. The mechanism of this stimulation of expression of the SERPINA3 protein and its function has not yet been understood. One existing hypothesis suggests the possibility of blocking serine proteases by the elevated levels of the SERPINA3 protein, which thus plays a chaperone protein role in prion formation [95]. Blocking the protease activity of the SERPINA3 protein in cultured nerve cells results in a reduction in prion accumulation [96,97]. This suggests that the increasing levels of the SERPINA3 protein in neuronal cells may promote the formation of prion plaque as well as other protein plaques, for example in AD [87,90], amyotrophic lateral sclerosis [98], and multiple system atrophy [99].

## 9. Conclusions

SERPINA3 is widely recognized as an acute-phase protein in the extracellular space when released into the blood plasma. Previous observations suggest that the glycosylation of the SERPINA3 protein determines its location and action, which has proven more complex (Figure 4). The dominant fraction with a high level of glycosylation is responsible for the antiprotease effect of this protein in the plasma. The most interesting observation, however, is that the weakly glycolyzed SERPINA3 protein in the nuclei of the cells may be a regulator of transcription. Most likely, this is achieved by binding DNA and increasing its condensation, thereby inhibiting DNA polymerase. In this way, SERPINA3 may be a factor in the inhibition of cell differentiation and division, preventing their initiation in the G0/G1 phase of the cell cycle. It seems that the SERPINA3 protein may play a similar role in the cellular antiviral response.

Observations on the appearance of the SERPINA3 protein in various pathological conditions of the human or model organism revealed its second nature. In cancer cells, the nuclear fraction of the SERPINA3 protein can activate the MAPK/ERK 1/2 and PI3κδ pathways by stimulating the NF-κβ signaling pathway and AKT phosphorylation and stopping the transition from the G2 phase to the M phase. This inhibits the transition of cells to apoptosis and stimulates tumor growth (Figure 4).

An additional, not fully understood aspect of the action of the SERPINA3 protein is its participation in the propagation of the formation of pathological deposits of proteins such as β-amyloid or prion protein, described in the cases of neurodegenerative diseases.

Although the receptor for the SERPINA3 protein is not known, it cannot be excluded that its release into the plasma in inflammation is aimed at delivering it to the cells involved in defense mechanisms, for example, B lymphocytes (own observation) to prevent their apoptosis too early.

The study of these mechanisms and the determination of the underlying factors that drive the changes in SERPINA3 protein glycosylation may create new diagnostic and therapeutic tools. In addition, it is necessary to confirm the diagnostic usefulness of the changes in the SERPINA3 protein levels observed in various diseases, since the prognostic potential of these changes seems to be very high, especially in the case of non-cancerous diseases.

## Figures and Tables

**Figure 1 biomedicines-11-00156-f001:**
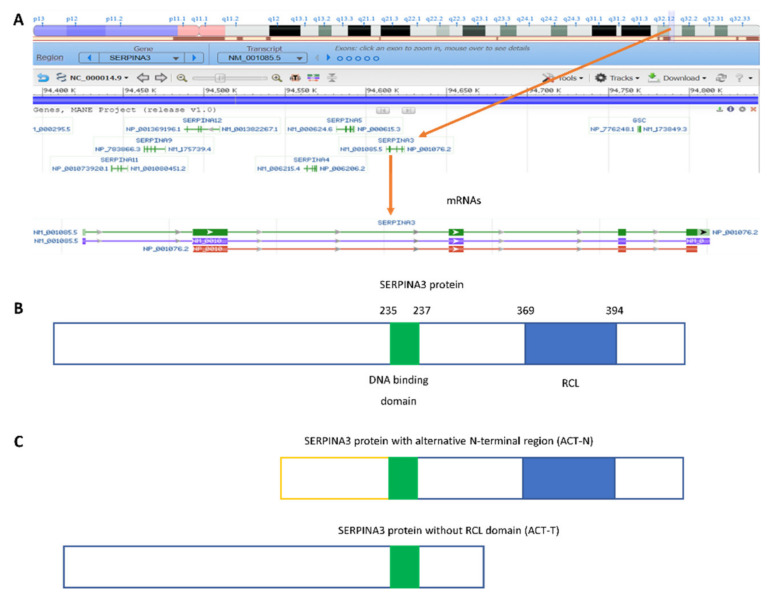
*SERPINA3* (serine protease inhibitor A3) gene: (**A**) chromosomal localization and canonical mRNA forms based on NCBI Genome data (https://www.ncbi.nlm.nih.gov/genome/gdv/browser/gene/?id=12, accessed on 21 December 2022); (**B**) scheme of canonical SERPINA3 protein; (**C**) protein products of alternative splicing: ACT-N (upper image) and ACT-T (lower image).

**Figure 2 biomedicines-11-00156-f002:**
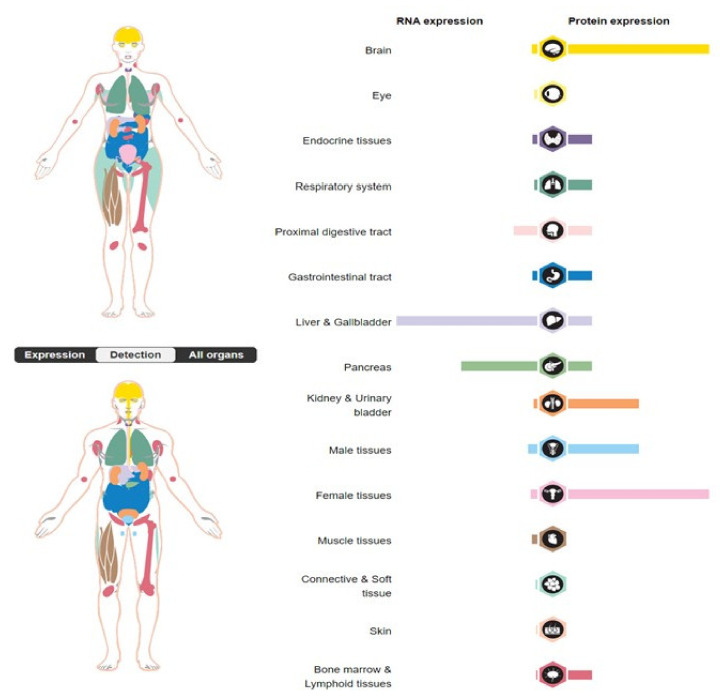
Tissue expression of the *SERPINA3* gene (https://www.proteinatlas.org/ENSG00000196136-SERPINA3/tissue; accessed on 17 December 2022).

**Figure 3 biomedicines-11-00156-f003:**
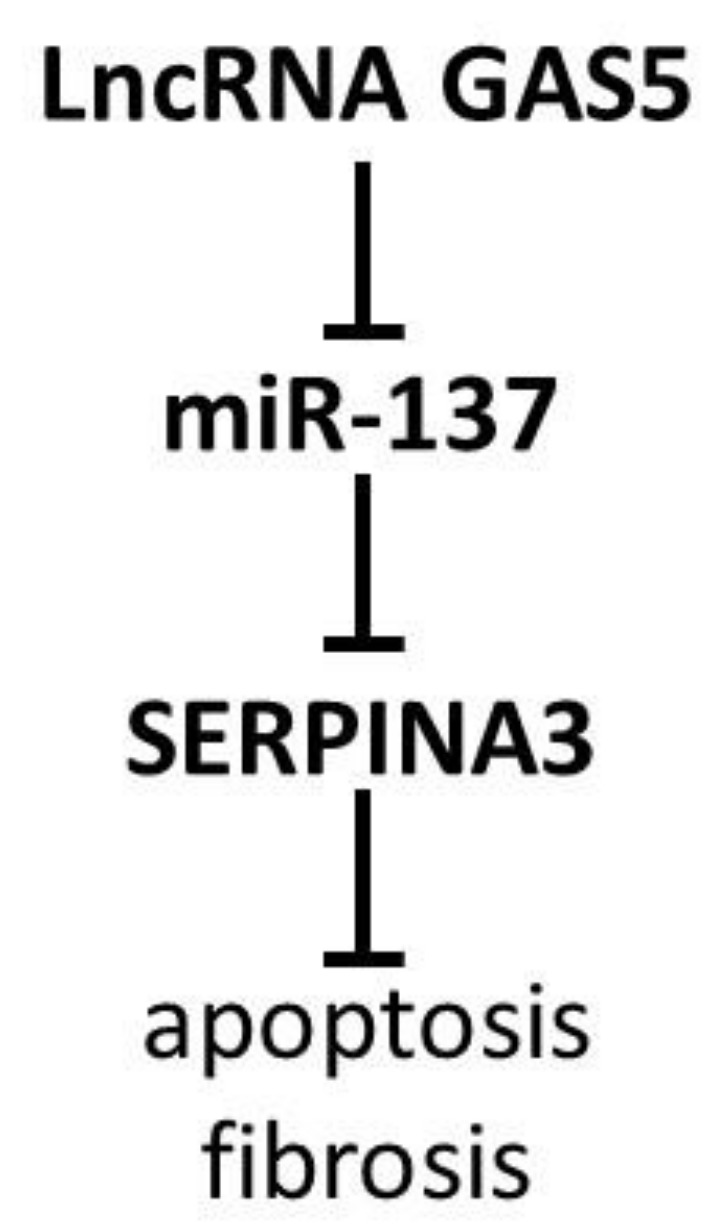
A decrease in SERPINA3 mRNA expression as a result of the inhibition of miR-137 by LncRNA GAS5 may cause lower apoptosis and/or fibrosis.

**Figure 4 biomedicines-11-00156-f004:**
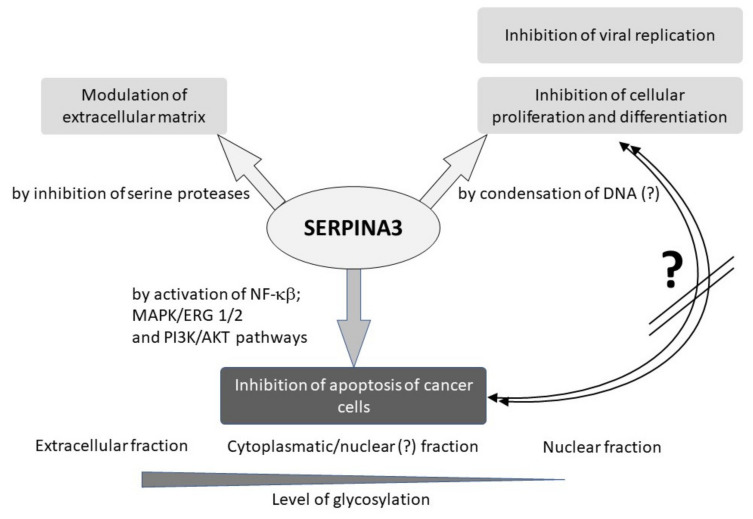
Possible role of SERPINA3 protein in cellular and extracellular processes.

## Data Availability

Data sharing not applicable.

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
