# Peer review of "SERPINA3: Stimulator or Inhibitor of Pathological Changes"

_biomedicines, 2023, doi:10.3390/biomedicines11010156_

Round 1

Reviewer 1 Report

The paper provided a comprehensive review of SERPINA3, including it's background, biological function,and related diseases. Also, the dicussion revealed a portrait of SERPINA3 in both pathological or physiological status of humen. Here I have a few questions and reconmmends.

1.Please use more figures in your paper, like the location of SERPINA3 in chromosome, the distribution of SERPINA3 in human body, and the subcellular location and so on. The figures could be more readable than only words.

2.Please notice that the non-coding RNA is not the most important related issue of a protein/gene, more information according to the central law about this protein/gene should be told.

3.Many researches were mentioned in the manuscript, but few of them can be connected together logically and the disscussion was not conclusive enough.

Reviewer 2 Report

The authors provide a brief literature survey on serpinA3 or a-1 antichymotrypsin. Long overshadowed by its more glamorous cousin a-1 antitrypsin, this inhibitor is enjoying increasing attention in the protease field in part due to its unusual role in modulating transcriptional activity. The review is quite brief and covers quite a lot of ground in relatively broad strokes. Several interesting stories are alluded to but not expanded on and there is generally little in the way of critical analysis. For instance, the murine ortholog of serpinA3 is a very different protein and yet there is scant discussion of this in the literature. Similarly, pathogen invasion and tissue damage seem to be important triggers for inhibitor activity and yet there is no over discussion of DAMP or PAMP signalling pathways or an in-depth discussion of neutrophil degranulation which is likely to be one of the most prolific sources of proteases relevant to SERPINA3.

Stylistically the review has numerous systematic errors. Crucially, the inhibitor’s activity is described as proteolytic in multiple locations rather than anti-proteolytic. Of lesser importance, lysine is present without its terminal E, Creutzfeldt -Jakob disease is misspelt, and there are numerous syntactical errors. For instance, SERPINA3 is described to have a “caring role” at line 335. The review would have been helped by more figures, especially showing the inhibitor’s structure annotated with relevant features (eg RCL and DBD domains). Of the figures provided, Figure 1 seems to add little to the review but Figure 2 is perhaps more useful. Whilst any collection of published ideas in an area is useful, I question the need for another SERPIN3A review after the recent publication in BBA by Soman and Nair. doi.org/10.1016/j.bbcan.2022.188760

Author Response

Thank you for your comments, which indicated the necessary corrections to our manuscript and allowed us to improve it. If possible, we made changes that included adding figures and supplementing information about, among others, the structure of the gene or mouse orthologus. Unfortunately, many studies on the action of the SERPINA3 protein in cells are performed in the context of its presence in cancer cells. As the reviewer rightly pointed out, this topic was presented by Soman and Nair. The problem is that in other states of the body, the processes in which the SERPINA3 protein seems to be involved are not so precisely studied, so they cannot be described as such, and at the same time the observations so far indicate the existence of significant differences in the action of SERPINA3 in different situations, which we tried to emphasize more strongly in the revised manuscript. The revised manuscript was linguistically corrected by the MDPI publishing house.

Round 2

Reviewer 2 Report

Much improved with nice figures. But there are still mentions of the serpin's proteolytic activity on page 4 at line 2 and 9. These should be removed. The serpin does not have proteolytic activity.

Author Response

Dear Reviewer,

Thank you for this comment, of course, I corrected the indicated fragments (you can see them below) so as to remove the erroneous information replacing it with the correct one:

The main function of SERPINA3 is the inhibition of serine proteases, such as chymotrypsins, cathepsin G, and mast cell chymase, by binding them in a stable complex, which prevents them from the proteolytic activity and consequently leads to changes in the extracellular matrix (ECM) [15–17].

Interestingly, such protein–DNA interaction does not affect the inhibitory activity of serine proteases [18].

Regards

Mateusz de Mezer